# Territorial Cooperation and Cross-Border Development: The Portuguese Dynamics

Pedro Chamusca

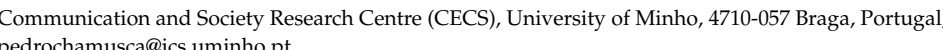

Communication and Society Research Centre (CECS), University of Minho, 4710-057 Braga, Portugal;
pedrochamusca@ics.uminho.pt

**Abstract:** This paper explores the effectiveness of cross-border cooperation programmes between Spain and Portugal, focusing on their impact and outcomes in Portuguese regions. Drawing on a comprehensive analysis of the programmes, the study examines the socio-economic dynamics in border regions, including job creation, population trends, and investment patterns. The research employs a mixed-methods approach, combining qualitative and quantitative data analysis. We argue that the cross-border territorial cooperation between Spain and Portugal has played a significant role in fostering regional development and addressing common challenges. While the concerted efforts have shown positive results in terms of economic growth and employment, they have not been sufficient to reverse the regressive demographic trends. Thus, it is essential to strengthen cooperation mechanisms, invest in human capital, and foster innovation so that the two countries can work together to create sustainable and inclusive development across their shared border regions.

**Keywords:** territorial cohesion; demographic dynamics; regional economy; EU funding

## 1. Introduction and Justification

Cross Border Cooperation (CBC) is a fundamental component of the European Union's (EU) policy towards its neighbouring countries. The CBC aims to achieve sustainable development along the external borders of the EU, as well as to address common challenges and reduce disparities in living standards across these borders. CBC fosters cooperation between EU countries and their neighbouring countries sharing a land border or sea crossing. The funding provided can support programmes between several EU and neighbourhood countries which, for example, are part of the same sea basin. The CBC is based on the EU's territorial cooperation model and is adapted to the specificities of external cooperation.

What sets CBC programmes apart from other forms of cooperation is the strong commitment and ownership of participating countries, which is based on several principles. These include a balanced partnership between participating countries, in which member states and neighbouring countries have an equal say in programme decisions, and projects receive funding only if implemented by partners on both sides of the border. Additionally, the management of the CBC programmes is entrusted to a local or national authority in a member state, which is jointly selected by all participating countries in the programme. A common legal framework and implementation rules have been established for CBC, which have been simplified and adapted based on previous experiences.

CBC has three primary objectives. First, it promotes economic and social development in border areas, which can lead to the growth of new businesses and industries. Second, it addresses common challenges related to the environment, public health, safety, and security. Finally, CBC aims to create better conditions for the mobility of people, goods, and capital across borders. These objectives are crucial for creating sustainable development and reducing disparities in living standards along the EU's external borders.

CBC is a vital tool for promoting cooperation and development between the EU and its neighbouring countries. Portugal and Spain share a border that stretches for more than

1200 km, and as such, the two countries have a long history of cross-border cooperation, which has evolved over the years to become an essential aspect of their bilateral relations.

The objective of this article is to provide an overview of the recent dynamics of cross-border cooperation between Portugal and Spain, analysing its main programmes and their impacts on the Portuguese border regions. The research questions guiding this study are: What is the recent context of cross-border cooperation between Portugal and Spain, and what have been the major drivers of this cooperation over the years? What are the main cross-border cooperation programmes between Portugal and Spain, and how have they evolved over time? Finally, what are the main impacts of cross-border cooperation on the economic and social aspects of the border regions?

To answer these research questions, the article is structured into six main sections, besides this introduction. Section 2 provides an overview of the concept of cross-border cooperation, its advantages, and the main principles of cross-border cooperation programmes in the EU. The Section 3 presents the research methodology. Section 4 presents the possible evaluation of two ongoing programmes of territorial cooperation in cross-border regions in Portugal. Section 5 discusses the research results, namely the impacts on the development of cross-border regions. Finally, the article presents the major conclusions and next steps.

## 2. Theoretical and Conceptual Background

### 2.1. Territorial Cooperation

Territorial cooperation has become a crucial element in regional development, and it refers to the collaboration between regions, cities, and other local actors across borders to achieve common goals and tackle shared challenges. This cooperation has become increasingly relevant in today's globalised world, where many of the challenges facing regions and cities are not confined by national borders. The idea behind territorial cooperation is that the problems of one region or city can be solved through collaboration with other regions or cities facing similar challenges.

In this context, cross-border cooperation has become an essential aspect of territorial cooperation that takes place between regions or cities located on either side of a border. The primary goal of cross-border cooperation is to address common challenges that transcend national boundaries, such as environmental protection, cultural exchange, or economic development (Brunet-Jailly 2022).

Cross-border cooperation aims to promote sustainable development and address the common challenges faced by border regions. It seeks to reduce disparities in living standards, improve the quality of life of citizens in border regions, and address common challenges such as environmental protection, cultural exchange, or economic development (European Commission 2021). Furthermore, cross-border cooperation helps to strengthen the relationships between regions and cities, which can lead to more comprehensive and sustainable cooperation in the future (Beer and Clower 2020; Decoville et al. 2013). The establishment of cross-border cooperation agreements and partnerships can help promote trust, mutual understanding, and collaboration between regions and cities, thus creating an environment that is conducive to regional integration and development.

### 2.2. Cooperating: Why and What for?

Territorial cooperation has become increasingly relevant in the field of regional development due to the many benefits it can bring to regions and cities. Cooperation allows regions and cities to address common challenges that they cannot solve alone by pooling their resources and expertise. This can lead to increased economic growth, improved social inclusion, and enhanced environmental protection (While et al. 2013). Additionally, territorial cooperation can foster learning and exchange of best practices, leading to the development of new ideas and solutions.

However, cooperation can also face several challenges that must be considered to ensure its success. One of the main risks is the existence of differences in administrative cultures, legal frameworks, and financial capacities between the participating regions and cities

(Gasparini 2014). These differences can lead to misunderstandings, miscommunications, and ultimately, the failure of the cooperation initiative (Table 1).

**Table 1.** Key figures on territorial cooperation.

| Advantages | Risks | Key Elements of Successful Territorial Cooperation |
|---|---|---|
| ⇒ Increased economic growth: By cooperating and pooling resources, regions and cities can achieve economies of scale, attract investment, and enhance competitiveness. ⇒ Improved social inclusion: Cooperation can lead to the development of social policies and programmes that promote equal access to services and opportunities and reduce inequalities. ⇒ Enhanced environmental protection: Cooperation can lead to the development of joint strategies and initiatives for environmental protection and sustainable development, such as the creation of protected areas or the promotion of renewable energy. ⇒ Learning and exchange of best practices: Cooperation can promote learning and exchange of experiences and good practices between regions and cities, leading to the development of new ideas and solutions. ⇒ Increased political stability: Cooperation can contribute to the development of shared interests and objectives and promote political stability and trust among different regions and cities. | ⇒ Differences in administrative cultures: Cooperation may face challenges due to differences in administrative cultures, norms, and values between different regions and cities. ⇒ Legal frameworks: Cooperation may be hindered by differences in legal frameworks and regulations between different regions and cities. ⇒ Financial capacities: Cooperation may face financial constraints, as different regions and cities may have different levels of resources and financial capacities. ⇒ Power imbalances: Cooperation may face power imbalances between different regions and cities, leading to unequal distribution of benefits and risks. ⇒ Lack of political will and commitment: Cooperation may face challenges due to the lack of political will and commitment among different regions and cities. | ⇒ Common vision and objectives: Successful territorial cooperation requires a clear understanding of common challenges and objectives and the definition of realistic and achievable goals. ⇒ Trust and communication: Successful territorial cooperation requires trust and effective communication mechanisms among different regions and cities. ⇒ Participation and ownership: Successful territorial cooperation requires the participation and ownership of all stakeholders, including local communities, civil society, and private sector actors. ⇒ Coordinated action and joint governance: Successful territorial cooperation requires coordinated action and joint governance mechanisms, such as joint management structures, decision-making processes, and monitoring systems. ⇒ Capacity building and learning: Successful territorial cooperation requires capacity building and learning mechanisms, such as the exchange of best practices, training programmes, and knowledge sharing platforms. |

Source: Own elaboration, based on: (Van Winsen 2009; Prokkola 2011; Faludi 2013; Allmendinger et al. 2014; Luukkonen and Moisio 2016; Chamusca et al. 2022; Chamusca 2023).

### 2.3. Cross-Border Cooperation in the EU

Cross-border cooperation within the European Union (EU) can be traced back to the 1970s when several European regions started to collaborate across national borders. However, it was not until the 1990s, with the Maastricht Treaty in 1992 and the establishment of the European Union's structural funds, that cross-border cooperation became an official EU policy objective. The structural funds aimed to promote economic and social cohesion among EU member states by supporting regional development initiatives, including cross-border cooperation projects.

In the early years, cross-border cooperation in the EU focused primarily on economic and infrastructure development, such as the construction of transport links and the establishment of industrial parks. However, with the signing of the Amsterdam Treaty in 1999, cross-border cooperation was given a broader remit, including social, environmental, and cultural aspects.

Over the years, cross-border cooperation has evolved from a focus on economic and infrastructure development to include social, environmental, and cultural cooperation. The

scope of cooperation has also expanded to include macro-regional strategies, which aim to promote cooperation and integration within a specific geographic area.

The major EU programmes for cross border cooperation include Interreg, Urbact, and the European Neighbourhood Instrument. These programmes were designed to promote economic, social, and territorial cohesion by funding cross-border projects in various areas, including innovation, energy, environment, and transport. One prominent example of EU-driven cross-border cooperation is the Interreg programme. Established in 1990, Interreg promotes collaboration across borders, fostering joint initiatives and projects between neighbouring regions. Through financial support, Interreg encourages the exchange of best practices, the development of shared solutions to common challenges, and the enhancement of economic and social ties. For instance, the Interreg Sudoe programme has facilitated cooperation among regions in Southern Europe, supporting projects in areas like renewable energy, research, and technological innovation (Barca 2009; Cappelli and Montobbio 2016).

Fostering innovation is a key focus of the EU's strategy for cross-border cooperation. Initiatives such as Urbact aim to encourage sustainable urban development by promoting the exchange of experiences and innovative solutions among cities across Europe. Urbact facilitates the creation of networks where cities can learn from each other's successes and challenges. An example is the Urbact Good Practice Transfer Network, which allows cities to share successful practices in areas like social inclusion, economic development, and environmental sustainability (Fauser 2019). The European Neighbourhood Instrument (ENI) is another critical tool for enhancing cooperation with neighbouring countries. While not exclusive to the EU, the ENI promotes stability, prosperity, and good governance in partner countries (Lange and Pires 2018). By investing in economic development, social progress, and infrastructure projects in these regions, the EU aims to create a more integrated and secure neighbourhood.

## 3. Methodology

This research is part of an official cross-border strategy development initiative, coordinated by the national governments of Portugal and Spain. Some data result from additional theoretical and empirical frameworks developed within this programme. In the context of this research, literature review and policy document analysis were used to understand recent dynamics and policy options related to territorial cooperation and, specifically, cross-border cooperation. The Portugal–Spain programmes were chosen as a case study, focusing on the impacts on the Portuguese regions. To analyse these territorial dynamics and effects, different indicators were collected, using data from the National Statistics Institute (INE) and the Portuguese Ministry of Territorial Cohesion's official sources. Some of this information was analysed using ArcMap 10.8.2., a GIS software from ESRI.

## 4. Results: Possible Evaluation of Cross-Border Regions' Programmes

### 4.1. European Territorial Cooperation from the Perspective of Portugal

Within the scope of European territorial cooperation, cross-border cooperation is particularly relevant, as emphasised by Commissioner Elisa Ferreira in October 2021 during the annual Interreg event. In fact, 37.5% of the EU population lives in areas close to country borders, along approximately 38 internal borders, characterized by geographical and linguistic barriers often associated with stigmas and various problems resulting from conflicts or other specific issues in people's lives. European cross-border cooperation supports cooperation between NUTS III regions of at least two Member States, directly focusing on border areas.

The establishment of cross-border cooperation strategies contributes to three fundamental objectives: (i) promoting economic prosperity through the use of distinctive resources within that territory; (ii) fostering balanced development of border regions, contributing to the social, economic, and territorial cohesion of the European space; (iii) strengthening the transition to sustainable processes of economic and social development, particularly through enhanced cooperation and greater efficiency in resource

allocation (EC 2010; Khmeleva et al. 2022). The more integrated phases of cross-border cooperation contribute to harmonious territorial development by retaining the most qualified individuals who would otherwise migrate to more economically and service-attractive national centres, typically located far from the border. This is achieved by enhancing the quality of life, including investments in innovation, healthcare, education, employment, and labour mobility.

In the case of the Spain–Portugal Cross-Border Cooperation Operational Programme (Tables 2 and 3), in line with the Interior Valorisation Programme, particularly its Axis 2 (Promoting cross-border cooperation for the internationalisation of goods and services), notable objectives include:

- Strengthening research.
- Technological development and innovation.
- Enhancing the competitiveness of small and medium-sized enterprises.
- Adaptation to climate change and prevention and management of risks.
- Preservation and protection of the environment.
- Promotion of resource efficiency.
- Increasing institutional capacity.
- Enhancing the efficiency of public administration.

**Table 2.** Key numbers of European Territorial Cooperation in Portugal (2022).

| Dimension | Key Numbers |
|---|---|
| Approved projects | 688 |
| Portuguese beneficiaries (partners) | 1539 |
| Million of ERDF from Portuguese partners (€) | 183.3 |
| Million of ERDF Programming for Portugal (indicative) (€) | 129.4 |
| Average ERDF aid per Portuguese partner (€) | 119,300 |
| Commitment rate for all ETC programmes in which Portugal participates (%) | 129 |
| Execution rate of funds related to Portuguese participation (%) | 71 |

Source: Ministry of Territorial Cohesion.

In the case of Portugal, for the next years, the European Commission approved the Spain–Portugal Cross-Border Cooperation Programme (POCTEP) in its Implementation Decision C 6125 of 22 August 2022, which is the largest cross-border cooperation programme in the European Union (EU) and will receive over €320 million in funding from the European Regional Development Fund (ERDF). Meanwhile, the POCTEP member states, in mutual collaboration and consensus as per the European Commission's guidelines, have been working for months on the preparation and monitoring of the POCTEP 2021–2027 activities. In this context, a working group consisting of the current national and regional authorities of POCTEP 2014–2020 was established in Seville with the aim of developing a common cross-border cooperation programme, defining priorities and common lines of action in accordance with the community legislation of the new programming period.

**Table 3.** Programming 2014–2020 and financial implementation of Cross-Border Cooperation Operational Programmes—Interreg A (2022).

| Cooperation Level | 2014–2020 Programming (Million Euros) | | | | Financial Indicators (Funds) regarding Portuguese Participation | | |
|---|---|---|---|---|---|---|---|
| | Total Investment | Public Expenditure | ERDF | ERDF for Portugal (Indicative) | Commitment Rate | Execution Rate | Achievement Rate |
| Cross-border | 868.1 | 849.1 | 653.7 | 95.1 | 118% | 62% | 52% |
| OkPOCTEP (Spain-Portugal Cross-Border Cooperation Programme) | 484.7 | 465.7 | 346.6 | - | 120% | 66% | 55% |
| PO Mediterranean Basin | 234.6 | 234.6 | 188.2 | - | 91% | 6% | 7% |
| PO MAC (Madeira-Azores-Canary Islands Cross-Border Cooperation Programme) | 148.8 | 148.8 | 118.9 | - | 114% | 51% | 44% |

| Cooperation Level | Total | | | Portuguese Participation | | | |
|---|---|---|---|---|---|---|---|
| | Approved Projects | Approved Projects with ERDF Funding (€M) | Overall Commitment Rate | Projects with Portuguese Partners | Portuguese Partners | ERDF Funding with Portuguese Partners (€M) | Average ERDF Support per Portuguese Partner (€M) |
| Cross-border | 445 | 656.9 | 100% | 342 | 995 | 112.6 | 113.21 |
| POCTEP | 238 | 345.0 | 100% | 238 | 717 | 84.9 | 118.41 |
| PO Mediterranean Basin | 80 | 186.9 | 97% | 3 | 3 | 0.5 | 181.80 |
| PO MAC | 127 | 125.0 | 105% | 101 | 275 | 27.2 | 98.91 |

Source: Ministry of Territorial Cohesion.

*4.2. Common Cross-Border Development Strategy (ECDT)*

The Common Cross-Border Development Strategy (ECDT) was approved between Portugal and Spain at the XXXI Luso–Spanish Summit, held in Guarda on 10 October 2020.

On the Portuguese side, the ECDT includes 145 municipalities, 1551 parishes (50.2% of the total Portuguese parishes), 57,138 km$^2$, and 1.6 million inhabitants. It encompasses the sub-regions (NUTS III) of Alto Minho, Cávado, Ave, Alto Tâmega, Tâmega e Sousa, Douro, Terras de Trás-os-Montes, Coimbra Region, Leiria Region, Viseu Dão-Lafões, Beira Baixa, Médio Tejo, Beiras, and Serra da Estrela, Baixo Alentejo, Alto Alentejo, Alentejo Central, and Algarve, representing 62% of Portugal's territory.

On the Spanish side, it includes 1231 municipalities, 86,561 km$^2$, and 3.3 million inhabitants. It encompasses all the municipalities in the border provinces (Badajoz, Cáceres, Huelva, Ourense, Pontevedra, Salamanca, and Zamora), accounting for a total of 17% of Spain's territory.

The strategy is based on a set of measures covering various thematic areas, focused on five axes, which are further divided into actions.

- Axis 1: Mobility, security, and elimination of contextual costs—12 actions
- Axis 2: Infrastructure and territorial connectivity—18 actions
- Axis 3: Joint management and sharing of services—20 actions
- Axis 4: Economic development and territorial innovation—13 actions
- Axis 5: Environment, energy, urban centres, and culture—17 actions

The analysis and evaluation of the implementation of the Common Cross-Border Development Strategy highlights the following points as essential measures:

i    The status of cross-border workers has been implemented. The Practical Guide for Cross-Border Work between Portugal and Spain was published, providing useful

information on all relevant topics for the implementation and facilitation of this measure, aiming to facilitate the mobility, access to information, and exercise of rights for workers who habitually reside or work in the cross-border territories of Portugal and Spain.

ii    The cross-border emergency number 112 has been implemented between Northern Portugal and Galicia. The cooperation protocol for the operationalisation of urgent medical assistance between Galicia and the Northern region of Portugal was signed on 14 December 2022. The protocol to extend this measure to the Castilla y León region (ES) and the Central region (PT) is being developed and will subsequently be extended to other cross-border regions in a phased manner.

iii    Innovation ecosystems: There are ongoing projects. A Memorandum of Understanding for the creation of the "Iberian FoodTec Lab" (IFL) was signed at the Luso–Spanish Summit (CLE), which will implement a collaborative Iberian agenda for research and innovation in the food sector, aimed at creating economic and social value in the Northern regions of Portugal, Galicia, and Castilla y León.

iv    Village Revitalisation: A Memorandum of Understanding is being prepared to be signed at the XXXIV Luso–Spanish Convention (CLE) with the aim of establishing a Working Group that will be responsible for defining the programmatic foundations for the development of an integrated revitalisation and promotion programme for villages.

v    Prevention of Domestic Violence and Violence against Women: A Memorandum of Understanding (MoU) was signed at the XXXIII CLE. At the end of December 2022, the Coordinating Commission of the MoU was established, with the priority of creating a Cross-Border Cooperation Network between organisations that support victims of violence against women and domestic violence during the first half of 2023.

vi    Multi-year Strategy for the Sustainability of Cross-Border Tourism and Common Cultural Agenda: A Memorandum of Understanding was signed at the XXXIII CLE, and in January 2023, coordinating and technical monitoring groups were established to work on the definition and implementation of the Common Cultural Agenda. The tourism ministries are also developing an action plan for the implementation of the strategy for the sustainability of cross-border tourism.

vii    REDCOT—Spain–Portugal Cross-Border Cooperation Network: The creation of this network is driven by the need to strengthen cooperation to address and resolve specific border issues, including emergency healthcare, the movement of cross-border workers, civil protection, mobility, infrastructure, depopulation, investment, goods transportation, and economic and social support, among others.

In the scope of the Recovery and Resilience Plan (PRR), investments are planned for five cross-border connections, totaling €110 million, which will bring these territories closer within the context of the Iberian Space and Europe in general. The implementation/improvement of these connections will contribute to greater cross-border territorial cohesion and the promotion of mobility in the inland regions. Additionally, it will reduce contextual costs in more peripheral territories, creating more attractive conditions for businesses and residents. This road connection consists of: Bragança–Puebla de Sanábria Connection (2021–2025 | €16 million); International Bridge over the Sever River (2021–2025 | €9 million); Alcoutim–Sanlúcar de Guadiana Bridge (2021–2025 | €9 million); EN103-Vinhais–Bragança construction of variant sections (2021–2025 | €31 million); and IC31 Connection—Castelo Branco–Monfortinho (2021–2026 | €45 million).

*4.3. Interior Valorisation Programme (PVI)*

Although it is still early to fully assess the effectiveness of the PVI, some implemented measures are already starting to bear fruit. For example, the creation of a more favourable tax regime for companies located in the interior has been well-received and has attracted new investments. Furthermore, investment in infrastructure has improved the quality

of life for people and has contributed to increasing the region's attractiveness for new investments and tourism.

However, the PVI still faces significant challenges, such as the lack of investment in certain sectors and difficulty in retaining people and businesses. The lack of opportunities for skilled employment and inadequate public services are still significant barriers. Therefore, it is necessary to continue evaluating and adjusting the programme to ensure that it fully achieves its objectives.

The Interior Valorisation Programme prioritises an implementation model of cross-cutting and multisectoral initiatives, known as +CO3SO (Constitute, Concretise, and Consolidate Synergies and Opportunities), involving different areas of governance and agents present in the territory. This model is structured into nine programmes that aggregate priority, relevant, and disruptive measures for the Interior territories (Table 4).

**Table 4.** Summary table of the review of the Interior Valorisation Plan.

| Axis | Programmes | Actions |
|---|---|---|
| Axis 1—Enhancing Endogenous Resources and Entrepreneurial Capacity of the Interior | +CO3SO Knowledge | Integrated strategy for the development of the interior based on knowledge transfer in co-creation environments |
| | | Mobility of students and researchers |
| | +CO3SO Digital | Integrated strategy for the development of the interior based on digital tools |
| | + CO3SO Social Innovation | Upscaling of social innovation in the interior |
| | | Social innovation in the interior in response to global challenges |
| | +CO3SO Tourism | Structuring of the offer of tourist products in the interior |
| | | Financial incentives for tourism in the interior |
| | | Promotion of tourism in the interior |
| | | Training in tourism in the interior |
| | +CO3SO Natural capital | Landscape, forests, and classified areas |
| | | Valorisation of resources and waste reduction |
| | | Enhancement of the potential of geological resources |
| Axis 2—Promoting Cross-Border Cooperation for the Internationalisation of Goods and Services | +CO3SO Border | Valorisation of the border in the interior territories |
| | | Resource sharing |
| | | Promotion of a favourable business environment in priority economic sectors |
| | +CO3SO Future Cooperation | Iberian Strategy for Territorial Cooperation |
| Axis 3—Attracting Investment and Retaining People in the Interior | +CO3SO Investment | National Programme for Investment Support in the Diaspora (PNAID) |
| | | Investment Attraction Programme for the Interior (PC2II) |
| | +CO3SO Employment | Employment in SMEs in the interior |
| | | Employment in social economy entities in the interior (IPSS—Social Entrepreneurship) |
| | | Digital Empowerment Programme for SMEs in the interior |
| | +CO3SO Benefits | Tax benefits in the interior |
| | | Improvement of Municipal Equipment (BEM Program) |
| | | Mobility to and within the interior |

Source: Resolution of the Council of Ministers No. 18/2020.

Four major goals or areas of intervention are assumed: people, companies, territory/border, and the scientific and technological system.

The major data available in the third quarterly report of 2022 show 2748 people relocated to Interior territories, 34,602 maintained/created jobs, 5432 trainees, and €6677.1 million invested in support measures, considering the investment in people (€1397.2 million), companies (€3660.3 million), the scientific and technological system (€198.3 million), and the territory (€1421.3 million). Based on this report, some data can be discussed, as follows.

### 4.3.1. Measures Aimed at People

- Mobility to the interior: 1313 approved applications, benefiting 2748 individuals, with an investment of €4.9 million carried out under the *Regressar and Emprego Interior* Mais programms.
- Tax benefits: Support provided to 742 families under the programmes targeting the attraction of students and families, resulting in a Fiscal Expense in Income Taxes of €139,109.66.
- Employment support/+CO3SO Employment: 1267 approved applications under the +CO3SO Social Entrepreneurship and +CO3SO Interior Employment programmes, creating 2546 jobs, with an approved investment of €129.4 million. The support already disbursed amounts to €24.2 million.
- Employment support/Integration of young people and/or adults in the labour market: 148 approved applications representing an investment of €8.5 million. The support already disbursed amounts to €3.2 million.
- Employment support/+ Highly Qualified Human Resources: 126 approved applications, creating 781 jobs, with an investment of €57.9 million. The support already disbursed amounts to €11.6 million.
- Employment support/SI2E (ESF): 1237 approved applications, creating 4164 jobs, with an investment of €14.9 million. The support already disbursed amounts to €6.3 million.
- Coworking/teleworking network: 89 participating municipalities, including 21 in the Northern region, 35 in the Central region, 19 in the Alentejo region, and 14 in the Algarve region. Currently, 65 spaces are in operation, providing 730 workstations. Approximately 200 workstations are permanently occupied.
- Proximity services: 1303 approved applications representing an investment of €964.2 million. This investment is distributed across education (564 applications/€379.8 million investment), health (76 applications/€300.1 million investment), culture and heritage (480 applications/€227.5 million investment), and public administration (183 applications/€56.5 million investment). The support already disbursed amounts to €350.6 million.
- Social inclusion: 382 approved applications representing an investment of €189.1 million. The support already disbursed amounts to €92.1 million.
- Social innovation: 669 approved applications representing an investment of €28.4 million. This investment is distributed among capacity building for social investment (201 applications/€2.1 million investment), partnerships for impact (458 applications/€25.1 million investment), and social impact bonds (10 applications/€1.3 million investment). The support already disbursed amounts to €9 million.

### 4.3.2. Measures Aimed at Companies

- Incentive system: 2926 approved applications, resulting in the creation of 20,181 jobs and an investment of €3013.1 million. This investment is distributed across R&D (232 applications/€99.3 million investment), productive innovation and entrepreneurship (1206 applications/€2661.7 million investment), internationalisation (754 applications/€209.3 million investment), and qualification (734 applications/€42.8 million investment). The support already disbursed amounts to €789.1 million.

- Support for collective actions: 122 approved applications, representing an investment of €67.2 million. This investment is distributed across internationalisation (43 applications/€28.2 million investment), qualification (49 applications/€20 million investment), and promotion of entrepreneurship (30 applications/€19 million investment). The support already disbursed amounts to €38.7 million.
- Entrepreneurship: 1764 approved applications, resulting in the creation of 994 jobs and an investment of €216.5 million. This investment is distributed across business incubators (10 applications/€3 million investment) and SI2E (1754 applications/€213.5 million investment). The support already disbursed amounts to €72.7 million.
- Support Programme for National Production: 686 approved applications, maintaining 5729 jobs and an investment of €80.8 million. Over 50% of the approved applications and support are in the Northern region. The support already disbursed amounts to €7.4 million.
- Tax benefits: Support provided to 28,626 companies under the programmes targeting the strengthening of the business fabric in the interior (27,305), reinvestment of profits (1121), attraction of investment (171), support for forest protection (25), and investment attraction in the forest sector (3), resulting in a Fiscal Expense in Income Taxes of €30,174,599.48.
- Business reception areas: 104 approved applications, representing an investment of €179.5 million. The support already disbursed amounts to €56.8 million. Under the Recovery and Resilience Plan (PRR), a maximum support amount of €73,004,751.46 is also planned to be granted.

### 4.3.3. Measures Aimed at the Technological System

- Support for science and technology: 180 approved applications, resulting in the creation of 207 jobs and an investment of €123.5 million. This investment is distributed across scientific infrastructure (8 applications/€11.5 million investment), technological infrastructure (26 applications/€51.3 million investment), scientific and technological research (108 applications/€47.5 million investment), knowledge transfer (33 applications/€12.2 million investment), and capacity building of Ciência Viva centres (5 applications/€0.99 million investment). The support already disbursed amounts to €49.4 million.
- Higher Education: 61 approved applications, involving 5432 trainees and an investment of €74.8 million. This investment is distributed across Technological Specialisation Courses (TeSP) (44 applications/€34.6 million investment), advanced training (4 applications/€4.1 million investment), training infrastructure (2 applications/€4.5 million investment), and +Superior programme (11 applications/€31.6 million investment). The support already disbursed amounts to €53.5 million.
- Collaborative Laboratories: Support provided to 10 collaborative laboratories, involving 206 human resources and an investment of €21.8 million.

### 4.3.4. Measures for the Territory/Border

- Environment: 256 approved applications, representing an investment of €86.1 million. This investment is distributed across areas such as energy efficiency in businesses (3 applications/€0.68 million investment), energy efficiency in housing (33 applications/€22.4 million investment), energy efficiency in public infrastructure (219 applications/€62.9 million investment), and air quality monitoring (1 application/€0.14 million investment). The support already disbursed amounts to €28.4 million.
- Urban development: 1144 approved applications, representing an investment of €700.3 million. This investment is distributed across areas such as sustainable urban mobility (+Access) (66 applications/€51.7 million investment), sustainable urban mobility (226 applications/€112 million investment), IFFRU (2 applications/€33.6 million investment), Citizen Shops (2 applications/€0.29 million investment), urban rehabilitation (750 applications/€460.4 million investment), socio-economic and physical re-

generation of disadvantaged communities (95 applications/€41.7 million investment), and flexible and on-demand transportation (3 applications/€0.64 million investment). The support already disbursed amounts to €320.1 million.

- Territory enhancement programs: 306 approved applications, representing an investment of €169.7 million. This investment is distributed across areas such as institutional capacity building in territorial and sectoral partnerships (36 applications/€14.2 million investment), socio-economic development at the local level (35 applications/€1.6 million investment), integrated interventions in vulnerable territories (33 applications/€12.2 million investment), and PROVERE (190 applications/€118.1 million investment). The support already disbursed amounts to €71.6 million.
- European territorial cooperation: 688 approved projects, representing an investment of €235.2 million.
- Cross-border connections envisaged in the Common Strategy for Cross-Border Development: €100 million in support.
- Multiple-Purpose Hydraulic Project of Crato: €100 million investment planned by the end of 2025.

## 5. Discussion: Is Cross-Border Cooperation Promoting Development of the Portuguese Regions?

Knowing that the evaluation of programmes and projects cannot be based solely on their implementation or the amounts they mobilise, it is premature to assess the impact of the previously mentioned strategy and programme measures. This is due, on one hand, to the fact that they are ongoing and, therefore, there are actions that are not yet completed (or even initiated), and on the other hand, even regarding what has been accomplished, it is not possible to conduct a detailed analysis of their concrete effects on the inland territories, particularly in terms of job creation and population retention, given the short time elapsed since their implementation. Nevertheless, it appears clear that there is an economic dynamic, job creation, and promotion of entrepreneurship, which is particularly noticeable in the inland territories, where it is expected that the efforts of the programmes and projects contribute.

Understandably, the coordinated action has not been able to alter the regressive demographic trend. The available indicators show the maintenance of a depopulation process, which, although transversal to almost the entire country, is particularly intense in the more inland regions, especially those farthest from cities.

Between 2011 and 2021, Portugal lost approximately 2% of its resident population and 11% of the population under 25 years of age, which now represents only 23.4% of the residents, compared to 25.7% in 2011 (Table 5; Figure 1). The twelve intermunicipal communities with border territories experienced a reduction of 4.7% in the resident population and 14.5% in the young population during the same period, but in some cases, these declines assumed more significant proportions, with overall population decreases exceeding 10% and young population declines exceeding 20%, as was the case in Alto Tâmega, Douro, Beiras, and Serra da Estrela.

These social and demographic dynamics are reflected in a progressive alteration of the resident population profile in both countries, particularly in border areas (Figures 2 and 3). Indeed, there is a continuous decrease in birth rate, with the lowest values recorded on the Spanish border with the northern region of Portugal. This process is linked to a dynamic of zero or negative natural growth in several regions, with the effects mitigated by a positive migratory dynamic, capable of attracting and settling new residents, often originating from Portuguese and Spanish-speaking countries, as well as from Africa or Eastern Europe. Despite this, aging is increasing at an increasingly intense pace, with territorial diversity (particularly relevant in border areas) and associated with a vicious circle of low density (Chamusca et al. 2022), leading to aging (Figure 4), a decrease in human capital, both public and private investment, and an increase in economic and social asymmetries.

**Table 5.** Population dynamics 2011–2021.

| Territorial Unit | Inhabitants Change (%) (2011–2021) | <25 Years Inhabitants Change (%) (2011–2021) |
|---|---|---|
| Portugal | −2.06 | −11.03 |
| Continental Portugal | −1.89 | −10.41 |
| Alto Minho | −5.53 | −17.25 |
| Cávado | 1.58 | −12.73 |
| Alto Tâmega | −10.51 | −26.54 |
| Douro | −10.37 | −24.91 |
| Terras de Trás-os-Montes | −8.71 | −19.68 |
| Beira Baixa | −9.31 | −15.79 |
| Beiras e Serra da Estrela | −10.76 | −22.23 |
| Baixo Alentejo | −9.32 | −13.06 |
| Alto Alentejo | −11.46 | −16.65 |
| Alentejo Central | −8.53 | −13.60 |
| Algarve | 3.65 | −3.46 |
| Total NUTS III with boarder | −4.66 | −14.46 |
| Total NUTS III without boarder | −1.35 | −10.16 |

Source: Own elaboration, based on data from INE.

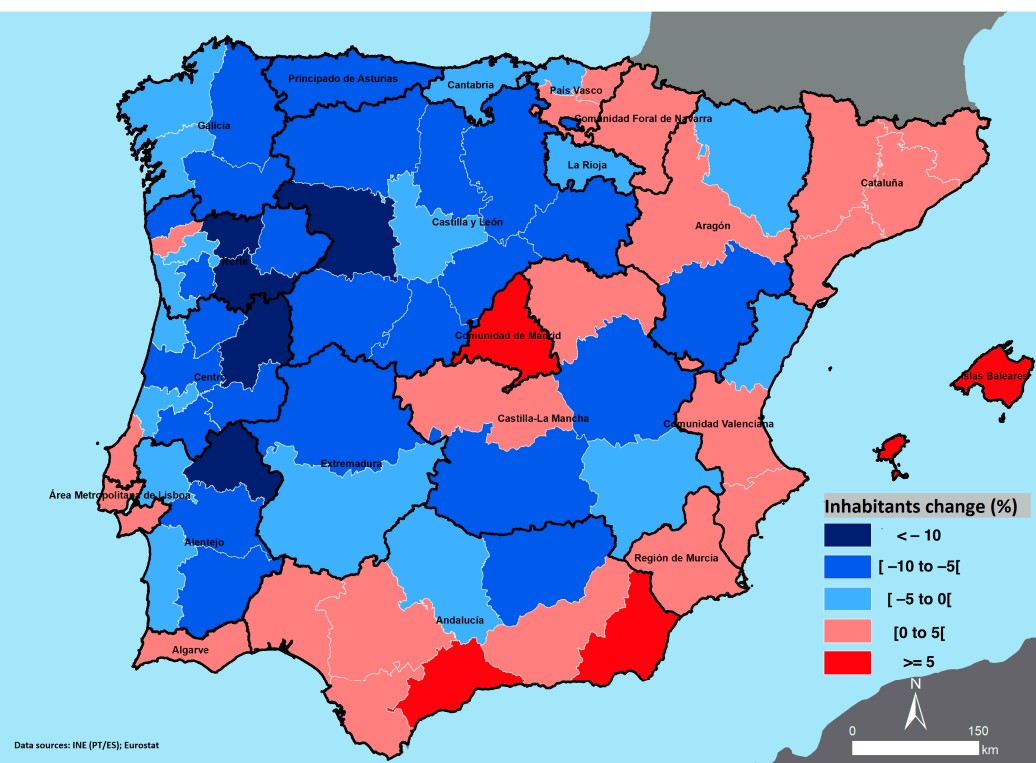

**Figure 1.** Variation of resident population (2011–2021). Source: Own elaboration, based on data from INE (Portugal and Spain).

In recent years, the border regions between Portugal and Spain have witnessed remarkable economic growth. This development is the result of significant investments in infrastructure, with a focus on improving transportation links and modernising energy networks. Additionally, cross-border cooperation initiatives have played a crucial role, facilitating collaboration between local entities, businesses, and organisations, promoting economic synergies and stimulating bilateral trade (Medeiros 2018; OECD 2022). Economic diversification has also been a priority, with efforts concentrated on promoting entrepreneurship and innovation, driving sectors such as technology, sustainable tourism, and agribusiness (Medeiros 2019).

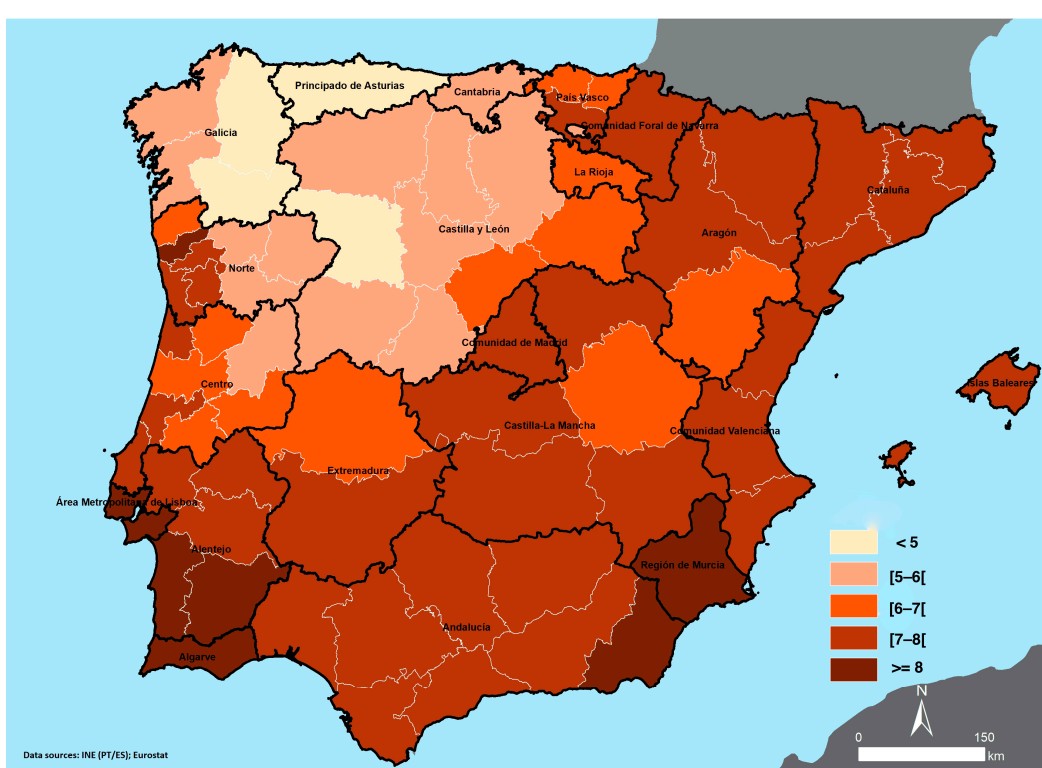

**Figure 2.** Birth rate (2021). Source: Own elaboration, based on data from INE (Portugal and Spain).

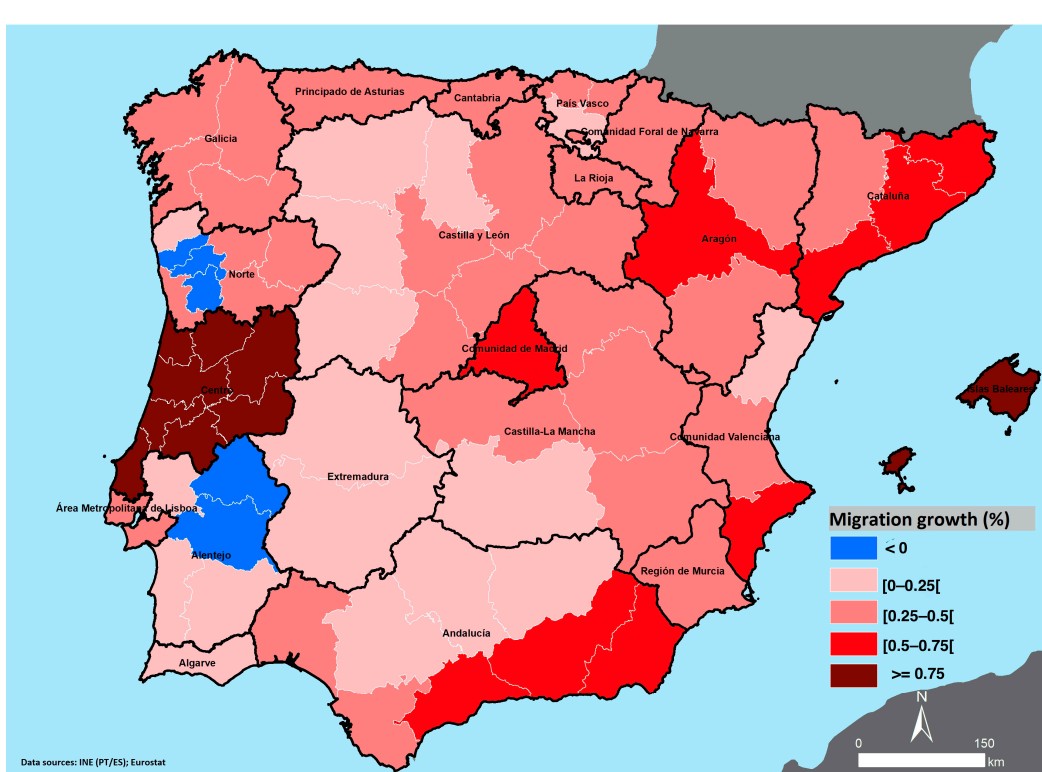

**Figure 3.** Migration rate (2021). Source: Own elaboration, based on data from INE (Portugal and Spain).

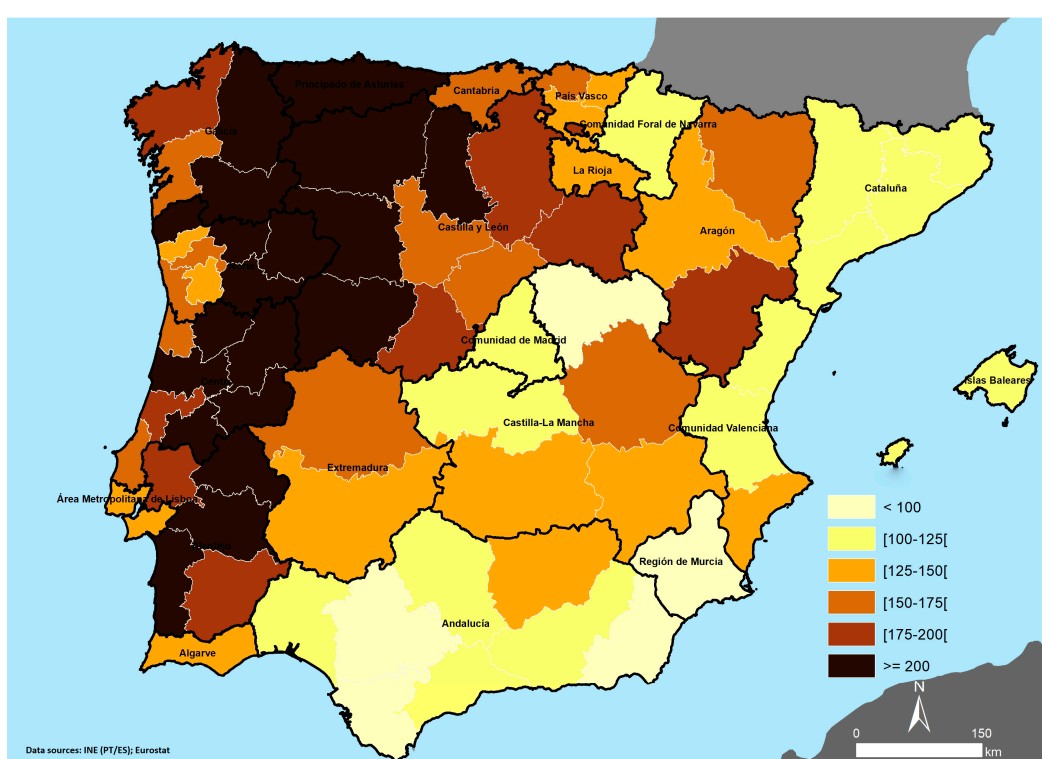

**Figure 4.** Aging index (2021). Source: Own elaboration, based on data from INE (Portugal and Spain).

Despite these advances, challenges persist, especially regarding demographic imbalances and social issues. Economic growth in border regions often faces obstacles related to population density and the need to retain local talent. Public policies have been directed at addressing these issues, encouraging investment in education, health, and social development programmes to ensure more inclusive and sustainable economic growth in the border areas between Portugal and Spain (Medeiros 2018; ESPON 2019).

In terms of job creation (Table 6), border areas have a performance closer to the national average, with a positive dynamic observed between 2016 and 2020, characterised by an increase in the number of companies (8.8% in Portugal and 7% in border areas) and job positions (5.2% compared to 4.2%).

**Table 6.** Employment dynamics 2016–2020.

| Territorial Unit | Change in Employees with a Dependent Employment (%) (2016–2020) | Variation in the Number of Companies (%) (2016–2020) |
|---|---|---|
| Portugal | 5.22 | 8.77 |
| Continental Portugal | 5.31 | 8.70 |
| Alto Minho | 11.62 | 5.13 |
| Cávado | 9.63 | 11.85 |
| Alto Tâmega | 8.26 | 6.29 |
| Douro | 7.03 | 2.87 |
| Terras de Trás-os-Montes | 3.27 | 7.64 |
| Beira Baixa | 5.24 | 2.52 |
| Beiras e Serra da Estrela | −0.96 | 3.69 |
| Baixo Alentejo | 1.12 | 6.46 |
| Alto Alentejo | 0.86 | 1.46 |
| Alentejo Central | 3.40 | 4.58 |
| Algarve | −1.89 | 9.90 |
| Total NUTS III with boarder | 4.16 | 6.98 |
| Total NUTS III without boarder | 5.46 | 9.32 |

Source: Own elaboration, based on data from INE.

This scenario is explained by the social and demographic dynamics mentioned earlier, especially the vicious circle of low density and the adopted strategy for its reversal. Despite this, and the very positive economic dynamics and job creation, we observe that business density and the weight of the industrial sector are still very incipient in border areas (Figures 5 and 6). Employment heavily relies on the tertiary sector, particularly in public employment within municipal, regional, or national entities or companies (Chamusca et al. 2022).

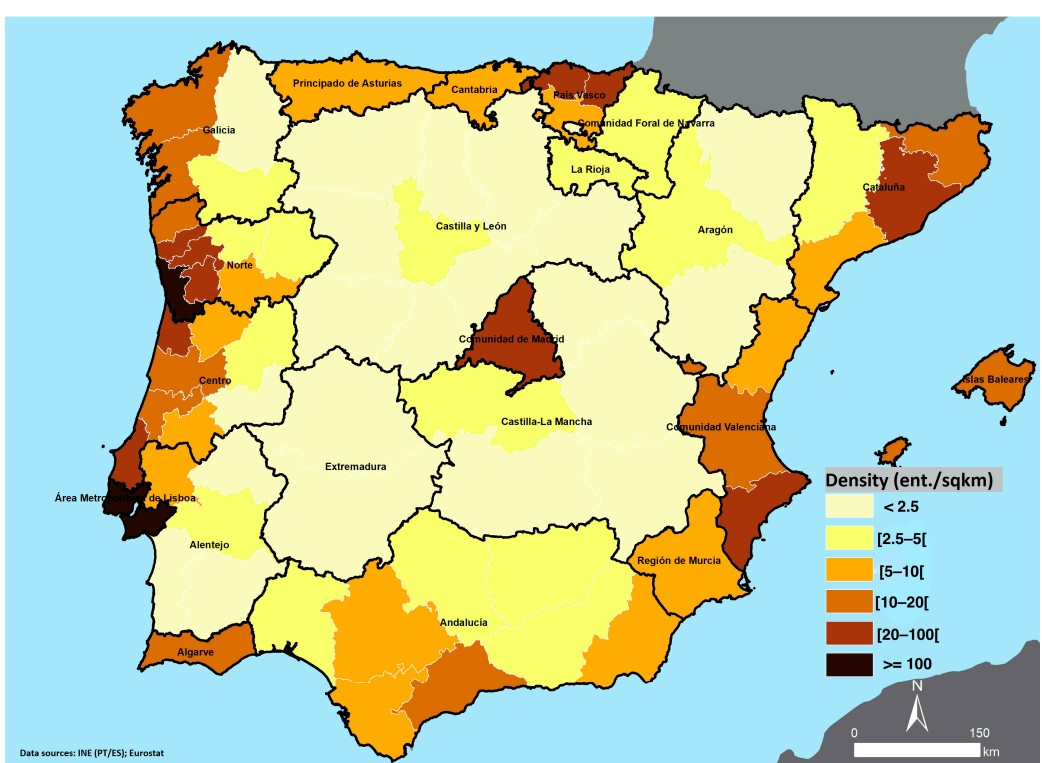

**Figure 5.** Density of enterprises (2021). Source: Own elaboration, based on data from INE (Portugal and Spain).

In summary, considering the successful implementation of public policies, it is important to consider national dynamics for a better assessment of programme effectiveness, to be carried out over a longer time horizon, and the additional challenges of reversing a regressive context in low-density areas, characterised primarily by four major issues: declining birth rate, aging population, loss of human capital, and investment difficulties.

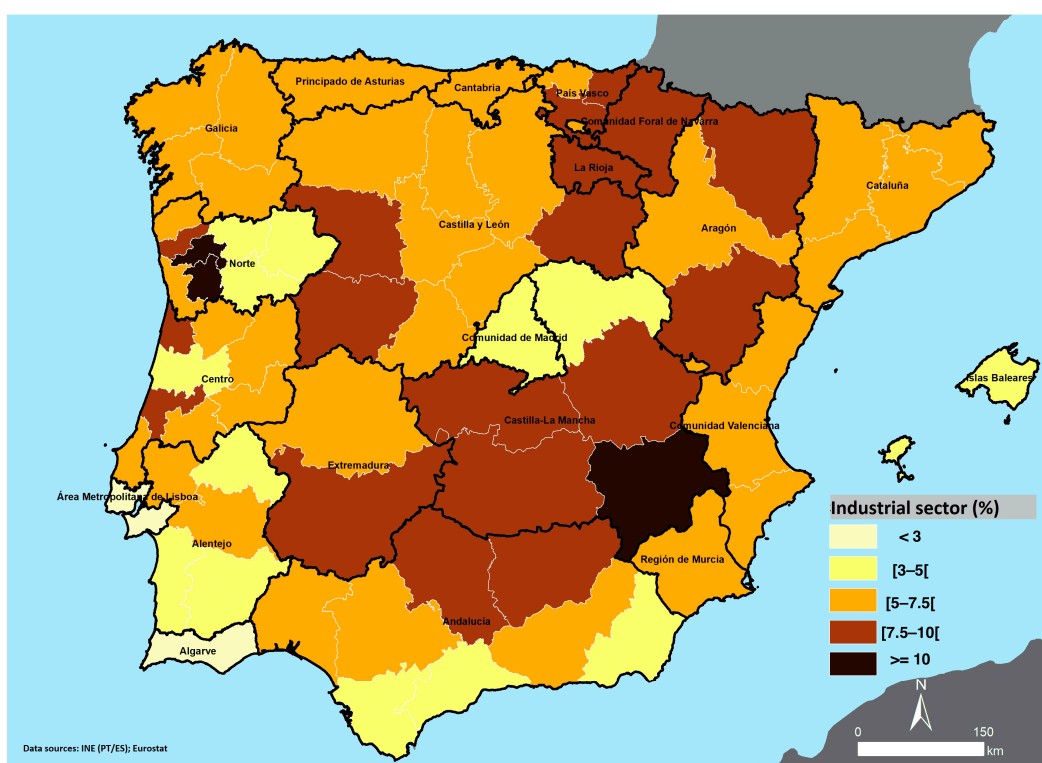

**Figure 6.** Relevance of industrial enterprises (2021). Source: Own elaboration, based on data from INE (Portugal and Spain).

## 6. Conclusions

The cross-border territorial cooperation between Spain and Portugal has played a significant role in fostering regional development and addressing common challenges. Throughout this text, we have discussed various measures and initiatives aimed at promoting cooperation in areas such as the economy, employment, technology, and the environment. Despite the ongoing nature of these programmes and the need for a more extensive evaluation, it is evident that they have contributed to economic dynamism, job creation, and entrepreneurship, particularly in the interior regions.

While concerted efforts have shown positive results in terms of economic growth and employment, they have not been sufficient to reverse the regressive demographic trends. The available indicators indicate a continued depopulation process, with more pronounced effects in the inland and remote regions, far from urban centres. Over the period from 2011 to 2021, Portugal experienced a population decline of approximately 2%, with a significant decrease of 11% in the population under 25 years old. The border communities faced even more substantial challenges, with population reductions exceeding 4.7% and a decline of over 14.5% in the young population, as observed in several intermunicipal communities such as Alto Tâmega, Douro, Beiras, and Serra da Estrela.

In terms of employment dynamics, the border areas have shown a performance close to the national average, exhibiting positive trends between 2016 and 2020. This period witnessed an increase in the number of companies by 8.8% in Portugal and 7% in the border areas, along with growth in job opportunities by 5.2% and 4.2%, respectively. These figures indicate a positive momentum in job creation and business development, contributing to the overall economic vitality of the cross-border regions.

To achieve more comprehensive and lasting results, it is crucial to consider the long-term national dynamics and address the underlying issues that hinder progress in low-density areas. Challenges such as declining birth rates, population aging, human capital loss, and investment difficulties need to be tackled strategically.

The cross-border cooperation between Spain and Portugal has demonstrated its importance in promoting regional development and addressing common challenges. While positive economic and employment dynamics have been observed, the significant demographic and structural obstacles in low-density areas call for sustained efforts and targeted interventions. By continuing to strengthen cooperation mechanisms, investing in human capital, and fostering innovation, the two countries can work together to create sustainable and inclusive development across their shared border regions.

## 7. Future Developments and Existing Knowledge

The conclusions drawn from the analysis of cross-border territorial cooperation between Spain and Portugal underscore the significant impact on regional development and shared challenge mitigation. These findings align with broader studies on transnational collaboration, where initiatives aimed at fostering economic growth, employment, and technological advancements have demonstrated positive outcomes. Cross-referencing these conclusions with international research reveals common trends in the positive influence of cooperation on economic dynamism and entrepreneurship in border regions.

While the success of concerted efforts is evident in economic indicators, addressing demographic challenges remains a shared struggle. Similar studies in various European contexts have noted persistent depopulation trends in remote areas, despite regional development initiatives. The observed decline in Portugal's population, especially among the youth in border communities, mirrors findings in other cross-border regions across Europe.

Examining employment dynamics reinforces the positive impact of cross-border cooperation on job creation and business development. The performance of border areas closely aligning with national averages, as seen in the growth of companies and job opportunities between 2016 and 2020, corresponds with the positive trends reported in transnational collaborative efforts elsewhere. This consistency supports the broader argument that cross-border cooperation can contribute significantly to the overall economic vitality of shared regions.

However, the study emphasises that sustained efforts are needed to overcome demographic and structural obstacles in low-density areas. This aligns with international research pointing to common challenges faced by border regions, including declining birth rates, population aging, human capital loss, and investment difficulties. Strategic interventions are crucial to address these underlying issues and ensure lasting positive impacts on regional development.

In conclusion, the cross-border cooperation between Spain and Portugal has proven instrumental in promoting regional development and addressing shared challenges. While positive economic and employment dynamics are evident, the study highlights the need for sustained efforts to overcome demographic and structural obstacles in low-density areas. By leveraging cooperation mechanisms, investing in human capital, and fostering innovation, the two countries can collaboratively work towards sustainable and inclusive development across their shared border regions, aligning with the broader international discourse on transnational collaboration.

As future policy recommendations and areas for further scientific exploration, considerations within the framework of an effort to coordinate all planned investments in the Interior (from European, national, and municipal funds) are vital. To avoid redundancy and promote efficiency maximisation in line with the principles of territorial governance, some priorities for reflection may include investment in infrastructure to address chronic issues in transportation, energy, water, and communications in the most remote and disadvantaged regions. Emphasis should be placed on enhancing digital connectivity and accessibility, fostering investments, and attracting businesses and residents. Additionally, there should be a promotion of entrepreneurship and innovation by encouraging the establishment of new businesses and innovation in disadvantaged regions through policies involving fiscal incentives, professional training, and capacity building, as well as technical and financial

support. Furthermore, valorisation of endogenous resources is essential, preserving and enhancing the value of natural, cultural, and heritage resources, while developing sustainable economic activities that generate income and create local employment. Support for agricultural development should be in place by incentivising the production and marketing of quality agricultural products. Finally, sustainable tourism development is crucial, acknowledging tourism as a crucial economic activity in disadvantaged regions. It should be developed sustainably, integrated with local communities, and respecting the region's natural, cultural, and heritage resources while preserving its identity and diversity.

**Funding:** This research was funded by Fundação para a Ciência e Tecnologia grant number UIBD/00736/2020. APC is paid by the author.

**Institutional Review Board Statement:** Not applicable.

**Informed Consent Statement:** Not applicable.

**Data Availability Statement:** Data used in the research is available at the Ministry for Territorial Cohesion documents and webpage.

**Conflicts of Interest:** The authors declare no conflict of interest.

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
