# Peer review of "Territorial Cooperation and Cross-Border Development: The Portuguese Dynamics"

_socsci, doi:10.3390/socsci13020108_

Round 1
Reviewer 1 Report
Comments and Suggestions for Authors
Comments are attached as a pdf file

Author Response
Dear Reviewer,
Please find point by point answer to your comments in the attached file.

Reviewer 2 Report
Comments and Suggestions for Authors
Dear Authors,
1. You should improve the spelling and edition of the text. For example, in the line 99 “ – key figures….”
2. Methodology . The work contains a number of qualitative analyzes. Although quantitative data is cited, no quantitative methods were used. Many methods can be used to analyze the development and cross -border cooperation: for example, correlation, regression, clusters and cluster analysis.
3. The map (Fig. 1.) Contains descriptions in language other than the language of the article (English). (In the line 435).
4. In the line 442-443 You wrote: “ Employment dynamics 2011-2021, but the data in the table 6. concerns 2016-2020. Why?
5. The article was submitted in the letter "Social Sciences" and concerns the development and socio-economic dynamics. Let the Authors consider supplementing the content with an explanation of the category of social development. Analysis of social dynamics boils down to the analysis of population tables and employment.
6. The work contains a lot of valuable information and conclusions but can be supplemented with quantitative analysis using statistics.
7. There is a lack of discussions of the results obtained with other scientific work.
I trust that after completing and making corrections it will be a very interesting job.
Best Regards
Author Response

(The authors gave the same response as above.)

Round 2
Reviewer 1 Report
Comments and Suggestions for Authors
The authors have made the corrections I suggested. I have no new comments. The paper is interesting, correctly presented and the data sources and bibliography are correct.
Reviewer 2 Report
Comments and Suggestions for Authors Thank You!